# Subtle Changes at the RBD/hACE2 Interface During SARS-CoV-2 Variant Evolution: A Molecular Dynamics Study

**DOI:** 10.3390/biom15040541

**Published:** 2025-04-07

**Authors:** Aria Gheeraert, Vincent Leroux, Dominique Mias-Lucquin, Yasaman Karami, Laurent Vuillon, Isaure Chauvot de Beauchêne, Marie-Dominique Devignes, Ivan Rivalta, Bernard Maigret, Laurent Chaloin

**Affiliations:** 1Laboratory of Mathematics (LAMA), CNRS, University of Savoie Mont Blanc, 73370 Le Bourget-du-Lac, France; aria.gheeraert@univ-savoie.fr (A.G.); laurent.vuillon@univ-smb.fr (L.V.); 2Dipartimento di Chimica Industriale “Toso Montanari”, Università di Bologna, Viale del Risorgimento, 40129 Bologna, Italy; i.rivalta@unibo.it; 3LORIA, CNRS, Inria, University of Lorraine, 54506 Vandoeuvre-lès-Nancy, France; vincent.leroux@loria.fr (V.L.); dominique.mias-lucquin@loria.fr (D.M.-L.); yasaman.karami@inria.fr (Y.K.); isaure.chauvot-de-beauchene@loria.fr (I.C.d.B.); marie-dominique.devignes@loria.fr (M.-D.D.); 4ENS, CNRS, Laboratoire de Chimie UMR 5182, 69364 Lyon, France; 5Institut de Recherche en Infectiologie de Montpellier (IRIM), CNRS, University of Montpellier, 34293 Montpellier, France

**Keywords:** SARS-CoV-2, RBD-hACE2 binding, molecular dynamics simulations, variant evolution, per-residue interaction

## Abstract

The SARS-CoV-2 Omicron variants show different behavior compared to the previous variants, especially with respect to the Delta variant, which promotes a lower morbidity despite being much more contagious. In this perspective, we performed molecular dynamics (MD) simulations of the different spike RBD/hACE2 complexes corresponding to the WT, Delta and four Omicron variants. Carrying out a comprehensive analysis of residue interactions within and between the two partners allowed us to draw the profile of each variant by using complementary methods (PairInt, hydrophobic potential, contact PCA). PairInt calculations highlighted the residues most involved in electrostatic interactions, which make a strong contribution to the binding with highly stable interactions between spike RBD and hACE2. Apolar contacts made a substantial and complementary contribution in Omicron with the detection of two hydrophobic patches. Contact networks and cross-correlation matrices were able to detect subtle changes at point mutations as the S375F mutation occurring in all Omicron variants, which is likely to confer an advantage in binding stability. This study brings new highlights on the dynamic binding of spike RBD to hACE2, which may explain the final persistence of Omicron over Delta.

## 1. Introduction

Since the beginning of the SARS-CoV-2 pandemic at the end of 2019, the virus mutations have been constantly challenging curative medical attempts, prevention of infections, and public health policies aiming to put a halt to the contagion. Despite the successful development of an innovative vaccine [1] having had a decisive impact on the first two fronts [2,3], the virus is still spreading. Its eradication, even locally, is considered highly unlikely [4,5]. Since 2022, the Omicron variants have emerged, quickly replacing all other strains worldwide, whatever the dominant one was locally, and opening a new era in the disease history [6]. The spike protein (S) [7] lining the surface of coronaviruses giving them a crowned-like shape is, since the beginning, the major target for pharmaceutical intervention against SARS-CoV-2, with all successful vaccine designs based on its immune recognition [8]. Spike, a class I viral fusion protein, was long known to be the main culprit for the coronaviruses’ initial infection process, as they hook to the mammalian host cells in the respiratory tract. Regarding lineage B betacoronaviruses like SARS-CoV and SARS-CoV-2, this is done primarily through its association to the transmembrane angiotensin-converting enzyme 2 (ACE2) [9,10]. Compared to previous variants of concern, Omicron clearly induces a lower morbidity and possesses a significantly higher contagiousness [11,12]. Notwithstanding other Omicron specificities [12,13], virulence attenuation (favorable since virus circulation is compromised if too many hosts die) might be greatly attributed to nsp6 mutations, while spike had an impressive number of immune-escape-promoting ones [14]. This explains how the virus gained a critical infectivity boost, significantly weakening spike recognition-related immune protection from past exposure and vaccines, thus allowing for rapid reinfections [15]. Such adaptation is not uncommon for viruses and was already suggested by the analysis of S evolution on pre-Omicron variants [16], yet the witnessed evolutionary jump with Omicron was outstanding. This was particularly concerning and eventually drove major changes in public health practices such as abandoning “zero Covid” policies [17] and recommending the administration of regular vaccine booster doses [15,18]. Since then, Omicron variants vigorously followed the same trajectory [19,20,21]. It is now accepted that accelerated post-Omicron S mutations are continuously maintaining a balance between improved immune escape and other viral efficiency requirements, the most obvious being sustained affinity for the human ACE2 (hACE2) target. This process may involve the S structure being remodeled upon acquiring additional resistance, subsequently altering structural stability, hACE2 binding, or global mechanistical features, in ways that are only partially understood and, thus, hardly predictable [22,23]. Therefore, while a deep understanding of S-hACE2 molecular mechanisms is of prime importance, probably more so since the advent of Omicron, it may be an increasingly difficult undertaking as time passes. As was the case prior to Omicron, numerous structural studies of the S-hACE2 complex have been conducted through X-Ray/Cryo-EM-derived models, despite such approaches mostly being directed toward S-antibody model structures, which is understandable given how antibody resistance is driving current S evolution and, thus, is the focal point for vaccine design. This publicly available [24] knowledge is essential for understanding how mutations affect the nature of S-hACE2 interactions at the molecular level [25,26,27]. However, while a clear picture of the mutation–function–structure–dynamics relationships in play [28] was previously seen, the speed of mutation clusters appearing since 2022 became hard to follow. Indeed, pre-Omicron major variants had, at most, 3 mutations in the region involved in hACE2 binding (Mu), compared to 15 with Omicron BA.1, up to 29 now with the latest emerging variant, Omicron KP.3.

An up-to-date summary (as of 2024-09) of major spike mutations in the Omicron family can be found in Appendix A. For comparison, the evolution of SARS-CoV-2 S during the first two years of the pandemic prior to Omicron appearance is shown in Appendix A, and a side-by-side comparison clearly showing how Omicron represented a breaking point in SARS-CoV-2 evolution regarding the S binding region is shown in Appendix A. Here, we focus on structural investigations of the S-hACE2 complex. Experimentally derived models, hopefully produced efficiently with the help of significant biotechnological progresses [29,30], have, in our opinion, a fundamental limitation. Indeed, they only correspond to static, artificially stabilized at low temperatures, snapshots of the biomolecular phenomena being investigated, dynamical processes by nature. This may not matter much when a particularly stable molecular association is depicted; however, it is now acknowledged that SARS-CoV-2 may evolve by trading S-hACE2 binding affinity to a certain extent for additional immune escape. In addition, a comparison of available Omicron S-hACE2 models already indicates significant mutation-related conformational plasticity [25]. Therefore, it seems interesting to investigate biomolecular structural dynamics here as we cannot rule out the existence of an “experimental gap” that may be filled by using the “computational microscope” [31]. In the particular context of the SARS-CoV-2 pandemic, this gap is rapidly being filled by deep mutational scanning studies on the Wuhan strain [32] and, more recently, on Omicron variants [33].

Molecular dynamics (MD) simulations have already proven their worth in S-focused SARS-CoV-2 research for comparing binding differences between previous variants [34,35,36] or for investigating the dynamics of S trimeric conformations [37] and the role of glycans [38] that appears more complex than simply cloaking the virus against antibody recognition. We previously conducted such an MD-based study on the S-hACE2 complex system for the original (wild-type, WT) SARS-CoV-2 strain as well as for its Alpha, Beta, Gamma and Delta variants, using long (multiple 1 µs trajectories) all-atom, explicit-solvent simulations and mostly perturbation network analyses [39]. In this follow-up study, we describe new 1.5 µs S-hACE2 unconstrained MD simulations from experimentally derived models, targeting WT, Delta and the Omicron BA.1, BA.2, BA.4/5 variants. We carried out a comprehensive analysis of residue interactions between the two protein partners, using a variety of complementary methods, precisely monitoring electrostatic and hydrophobic interactions as well as contact networks over the course of the simulations. Instead of computing binding energies, we aim to provide qualitative insight on the systems dynamics, complementing existing structural knowledge. Nevertheless, it was interesting to compare the PairInt method with standard procedures used for computing binding free energies such as MM/GBSA [40] and their respective performances for detecting subtle or transient interactions. Altogether, this work may contribute to a better understanding of the structural changes that occurred during the transition from Delta to Omicron and between the first Omicron variants of concern. It also aims to further demonstrate that MD simulation is a useful addition to the strongly interdisciplinary methodological toolbox required to understand the threatening level of SARS-CoV-2 variants during their evolution.

## 2. Materials and Methods

### 2.1. Molecular Dynamics (MD) Simulations

A 3D model of the Delta RBD/hACE2 complex has been produced in a previous work via homology to the 6M0J PDB structure [39,41]. This structure includes the RBD from the original Wuhan SARS-CoV-2 strain (Wuhan/Hu-1 or 19A), hereafter called WT. We initially modeled the RBD/hACE2 complex for Omicron BA.1 according to early reports, erroneously bearing the mutation Q493K (marked hereafter as BA.1^+Q493K^) using the same procedure (given that there was no better structural starting point available at that time). BA.1 and BA.2 variants (with the proper Q493R mutation) were modeled using the more appropriate PDB structure 7T9L [26]. Eventually, BA.4 (also representative of BA.5, which shares the same RBD sequence) was derived from 7ZF7 [42] (Table 1 and Appendix A).

The preparation of coordinates, structures and input files for MD was carried out as previously described [39]. The spike RBD and hACE2 components of each complex contain 229 and 603 residues, respectively. Briefly, each RBD/hACE2 complex was parameterized using the CHARMM36m force field and then immersed in a water box (TIP3P water model and box size of 150 Å3) at a physiological salt concentration ([NaCl] = 0.154 M), and electrostatic neutrality was achieved by completing with few additional Na^+^ or Cl^−^ ions depending on the system total charge. Periodic boundary conditions in conjunction with particle mesh Ewald method were set up for each system. After energy minimization (64,000 steps of conjugate gradients) and 10 ns of equilibration, all simulations were carried out in the isobaric–isothermal ensemble, at constant temperature (300 K) and pressure (1 atm), using Langevin dynamics and Langevin piston as implemented in NAMD (v3.0α9 [50]). We recorded 15,000 frames from the production trajectory (1.5 µs; time step of 100 ps) for further analysis. MD trajectories were analyzed from 0 to 1.5 µs, except for contact Principal Component Analysis (cPCA) that was performed from 800 ns to 1500 ns based on the stability evaluation of PC values. As initial starting states may induce specific conformational behavior, three simulation replicas were run for the WT RBD/hACE2 complex (for both non-glycosylated and fully glycosylated systems) by restarting the simulation after equilibration by reinitializing the initial velocities. All the complexes were initially modeled by omitting the few glycosylations present on spike RBD and hACE2, which may contribute to the binding affinity of both partners. To evaluate the impact of these glycans, we modelled the WT system with the N-linked glycan on Asn343 for RBD and the 8 glycosylations described for hACE2 (7 N-linked glycans on Asn residues at positions 53, 90, 103, 322, 432, 546, 690 and the O-linked glycan on Thr730). The components in monosaccharides were selected according to the site-specific glycan analysis published by Newby et al. [51], and the modeling was achieved using the Glycan modeler module integrated in Charmm-GUI [52].

### 2.2. PCA and RMSD Calculations

Principal Component Analysis (PCA) based on 3D coordinates from MD simulations was achieved using a homemade Python script based of the ProDY package (v2.2.0), allowing for Essential Dynamics Analysis (EDA) [53], which was applied to the different complexes by using all backbone atoms (RBD and hACE2) after superposition. Root Mean Square Deviations (RMSD) were computed on superposed backbone atoms of the RBD component, in order to evaluate the conformational space visited during the simulation and to estimate the convergence of the simulations for each variant. Further, 2D RMSD plots were obtained using the MDAnalysis python package (v2.2.0) and its provided algorithms and modules [54,55,56].

### 2.3. Pair Interactions (PairInt) Analyses

For analyzing interactions between biomolecules in a molecular mechanics-based atomic model, we developed an approach referred to as pair interactions protocol [57,58]. This is based on a feature of NAMD package [50], allowing the user to change the standard output of an MD simulation so that the potential energy fluctuations between two non-bonded user-defined atom groups (referred to as a pair) are returned instead of the whole system energy. Reprocessing a pre-calculated MD trajectory in a dummy calculation (T = 0 K) can be achieved by using the built-in NAMD Tcl interpreter in order to obtain such data efficiently. However, our recent implementation of PairInt calculations represented a significant improvement over this hack and is also more flexible than alternatives such as the NAMD-Energy module from VMD (v1.9.4) [59]. The whole process consists of four successive stages: (1) generate a pseudo-PDB file containing pairs definitions as indexes, allowing granular PairInt calculations at various granularities (i.e., protein–protein, protein A residues–whole protein B, residue–residue); (2) create batches of NAMD configuration files for PairInt calculations (i.e., prepare all input data for computing the interactions between all individual residues from a protein and its target receptor or partner in a single command); (3) distribute the calculations on multiple CPU nodes (a single PairInt calculation uses a single node since no motion is actually computed by the parallelized MD engine) and extract the data from output; (4) optionally, prepare multiple plots for graphical representation (e.g., use of LaTeX templates).

Main advantages of the PairInt protocol are its adaptability to static models (e.g., X-ray-derived) up to large MD trajectories (>1,000,000 frames) and its independence to both the system equilibrium state and the nature of its constituents. Its major weakness is the dependence on the underlying molecular mechanics force field, namely here the CHARMM36m one [60]. Such a force field is explicitly parameterized against interactions with the TIP3P water model, which features high partial charges, and is, therefore, only well suited to explicit-solvent simulations. This is counterbalanced by high partial charges on all polar residues and fine-tuning other coupled parameters. As a result, most electrostatic forces are grossly overestimated individually, especially when salt bridges are involved. On the contrary, hydrophobic contacts between two groups are not directly translated as non-bonded interactions in the potential but rather indirectly through avoidance of destabilizing entropic fluctuations due to solvent exposure. To summarize, PairInt values represent the enthalpic part of non-bonded interactions, as explicitly defined through the force field parameters and seen by the MD engine. They typically allow for useful qualitative comparisons if their limitations are considered, especially upon comparing similar systems (e.g., different ligands bound to the same receptor, or, as in the present case, protein–protein systems differing by mutations in one of the partners).

All PairInt calculations were performed by launching Bash scripts that called NAMD and initially required three input files: an MD trajectory (DCD format), a topology file (PSF format), and force field parameters. The parameters for computing the potentials are strictly identical to those used during the simulations, except the cutoff distance. In the MD simulations, it is set at 10 Å with a linear decrease to 8 Å (switch distance from which non-bonded interactions are not accounted); when reprocessed, the cutoff was reduced to 6 Å (with a switch distance of 4 Å), which was done in order to eliminate longer-range electrostatic effects that may be spotted at distances where no direct interaction is actually happening. Such artifacts generally disappear in interactions between two large segments but might be misleading when considering single residues. In the present study, all PairInt calculations were performed on a 2013 MacBook Air (approximate performance: 1000 MD frames/minute for each atom pair on ~12,000-atom trajectories). For the analysis and a better clarity of the figures, MD trajectories were resampled with 100 ps frequency (15,000 frames for 1.5 µs, ~2.2 GB per trajectory) and stripped from solvent atoms, except for the complete analysis presented in Appendix A (100 ps frequency with solvent).

### 2.4. Computing Binding Free Energy Using MM/GBSA Model

The binding free energy was calculated using the Molecular Mechanics/General Born Surface Area (MM/GBSA) module developed by Valdés-Tresanco [61] using the single-trajectory protocol. For each hACE2/RBD complex, the hACE2 was treated as the receptor, and RBD was considered as the ligand. Prior to computing the energy, all trajectories produced by NAMD were transformed using cpptraj (AmberTools20) and the Amber ff14SB force field. To estimate the binding free energy of the receptor–ligand complex, we performed MM/GBSA calculations in triplicate on segments of 100–200 ns of the simulation (in the range from 800 ns to 1500 ns) by using the generalized Born model gb8, a salt concentration of 0.15 M (same as used for simulation in explicit water) and an internal dielectric constant of 2.5, as described for highly electrostatic interactions [62], leading to more realistic values in respect to experimental binding affinities. The entropic contribution was estimated by computing the interaction entropy on the same part of the trajectory at 298 K. Per-residue energy decomposition was achieved using a cutoff distance of 4 Å between receptor and ligand residues.

### 2.5. Hydrophobic Interactions Analysis

Molecular Hydrophobic Potential (MHP) was computed for the two partners (RBD and hACE2) present in each complex for each of the 15,000 MD frames using PLATINUM (v1.0) stand-alone software [63] and Ghose atomic hydrophobicity constants [64]. PLATINUM is a widely used software for analyzing hydrophobic interactions in protein–ligand or protein–protein interaction studies. An atom is considered hydrophobic if its computed MHP is strictly positive. Hydrophobic interactions between proteins are only considered when one hydrophobic atom of one partner is found closer than 4.2 Å to a hydrophobic atom of the second partner [65]. This distance is derived from the distribution of inter-atomic distances between hydrophobic atoms and roughly corresponds to the distance between two hydrophobic atoms (r~1.0 Å), between which a water molecule (d H_2_O~2.8 Å) cannot fit (required distance ~4.8 Å).

### 2.6. Contact Principal Component Analysis (cPCA)

For further analysis, we gather the contact weights of different frames in a matrix C of size $N_{frames}\times N_{contacts}$, as previously described [39]. Briefly, for contact PCA (cPCA), the scikit-learn implementation of PCA decomposition allowed us to extract the k-first principal components (PCs). Each PC is of size $N_{frames}$ and represents the projection of the frames on this component. During the decomposition, the (ordered) eigenvectors of the contact covariance matrix (size $N_{contacts}\times N_{contacts}$) are computed. Each of these eigenvectors corresponds to a principal component and is of size $N_{contacts}$. Thus, a PC contact network i (PCNi) can be associated with each eigenvector of rank i as a graph, in which the nodes represent the amino acids, the edges the contacts between pairs of residues, and the weights the values assigned to the contacts in this eigenvector of rank i. To each eigenvector corresponds an eigenvalue, which is representative of the importance of the principal component. In PCA, the eigenvalues and eigenvectors are ordered so that the PCs decrease importance with the component number.

Restricting to one or two given eigenvectors i and j (PC_i_, PC_j_), one can obtain a projected 2D free energy landscape of the system along the eigenvector dimensions, namely a bi-plot:$$\Delta G\left(PC_{i},PC_{j}\right)=-k_{B}T\left[P\left(PC_{i},PC_{j}\right)-G_{0}\right]$$
where $P\left(PC_{i},PC_{j}\right)$ is the probability distribution obtained from a bivariate kernel density estimate computed on the MD frames and $G_{0}$ is the free energy of the most probable state. This procedure is similar to the one described previously based on dihedral PCA to obtain a free energy landscape but uses different internal coordinates [39,66]. In fact, the distribution of frames in the PC_i_/PC_j_ subspace corresponds to a 2D representation of the free energy surface. The shape of this surface remains consistent across both the abstract units of PCA and energy units, due to the application of a linear transformation between the two. In this study, we consistently represent the distribution of frames using the abstract units of PCA. As stated in the Results section, a specific simulation time range was selected for cPCA according to the convergence and stability of the two first PCs. PC1/2 values showed a high stability between 800 ns and the end of the simulation (1500 ns). Therefore, this range was used for all systems, as similarly described in our previously published study on former variants [39]. To illustrate the contact networks corresponding to each PC (or eigenvector) around the point mutations, edges colored in red indicate stronger contact in frames with positive values and lower contact in frames with negative PC values, and vice versa for edges in blue. The edge width is proportional to the influence of the contact in the eigenvector (threshold applied of 0.04 or 4% of the eigenvector contribution).

### 2.7. Cross-Correlation Analysis and Linear Mutual Information

Normalized linear mutual information (nLMI) is a method for quantifying correlations between pairs of amino acids along MD trajectories [67]. The values of this metric range between 0 (no correlation) and 1 (highest correlation). In this study, we used the correlationplus program (v0.2.1) [68] to calculate the nLMI on the WT and variant (Delta, BA.1^+Q493K^, BA.1, BA.2, and BA.4) RBD/hACE2 complexes. From the obtained matrices of nLMI, we then analyzed the correlations within the RBD, within the hACE2 and between the two partners (RBD and hACE2). For every system, we used the 1.5 µs MD simulation trajectories as inputs for the correlationplus program. We calculated the nLMI between residues using the “calculate” module and analyzed the correlations higher than 0.75 between pairs of amino acids that are within 10 Å distance using the “visualize” module. This module builds a dynamic network where nodes correspond to c-alpha atoms [69,70,71]. Then, for nLMI values higher than a given threshold, bidirectional edges are added to construct the network. In parallel, we obtained the eigenvector centrality for correlations that are higher than 0.10 and within 200 Å distance (using the “analyze” module). This metric highlights the importance of a given node by considering its interactions with its highly connected neighbors [72,73]. The PyMol Molecular Graphics System (v2.5, Schrödinger, LLC, New-York, NY, USA) was used to visualize the proteins and map the eigenvector centrality values on the structure.

## 3. Results

### 3.1. MD Simulations, Conformational Sampling and Dynamic Behaviors

The non-supervised Principal Component Analysis (PCA) method has been extensively used for analyzing MD simulations in order to extract essential dynamics that describe main or relevant conformational motions [74]. For starting our analysis, we first checked the conformational sampling and evolution of the MD trajectories obtained with the different RBD/hACE2 complexes by analyzing the essential dynamics described by the two first PCs (see Materials and Methods for detailed descriptions of the 3D models). The corresponding 2D projections of this regular PCA are presented in Figure 1 (note that the two first PCs explained the variance only by about 50% and cannot serve as a definitive interpretation, see Appendix A). Overall, the conformational space visited by all systems is similar, as shown in the overlay of the five variants and WT (Figure 1B). However, some differences appear upon close examination of individual scatterplots (Figure 1A).

For WT, the dynamics are mainly represented by the first principal component (PC1) and not always correlated with PC2, likely meaning less correlated dynamics motions. A visual inspection indicated that the largest motions were localized at some extremities of RBD or hACE2, except one loop (K444-G447) in the vicinity of the binding interface (not involved in direct interactions with hACE2 but close to residue Q498). The prevalence of this component (PC1) was also confirmed by analyzing the three replicas in the presence or in the absence of glycans (Figure 1B), with PC1 values ranging from −6 to +4. The presence of glycans does not seem to change the conformational space visited along PC2 (values from −4 to +2). At first glance, all the investigated variants, except BA.1^+Q493K^, which behaves similarly to WT, display a larger exploration of the conformational space, as shown by the amplitude of PC2, suggesting more dynamics motions or more conformational transitions visited. When comparing Delta and Omicron variants, the visited space has changed and appears more concentrated for Omicron (especially for BA.4, showing a shorter amplitude in PC1), suggesting a more constrained essential dynamic. However, this regular PCA is not sufficient to infer the global dynamics of each complex, and it is difficult to compare between them, since the visited space can be slightly variable in size from one replica to another, as shown for WT (at least for PC2, as shown in Figure 1B). The role of glycans (only one linked to the spike RBD domain at position 343 and 6 for hACE2) remains modest, with a trend towards a smaller conformational space exploration in PC2.

This preliminary analysis is supported by the calculated 2D RMSD matrix plots for the RBD component of each system (Figure 2). Firstly, the conformational behavior of the WT RBD in the RBD/hACE2 complex is different from the ones obtained with the variants. In WT, the RBD structure weakly fluctuates around a single conformational state near the X-ray starting structure during the first 500 ns while being less stable after this time without presenting a clear stable state afterwards.

The BA.1 variant behaves more like WT than other variants, with a short stable period around the starting structure (first 250 ns), followed by unstable events corresponding to conformational states visited several times. The situation is changing for Delta, BA.1^+Q493K^, BA.2 and BA.4 variants, for which the RBD presents two main conformational states. Indeed, the exploration space seems to evolve from the unstable states displayed by the WT and the BA.1 variant to more stable conformations in the Delta and BA.1^+Q493K^ variants, up to even more stable conformations observed during long periods of MD simulation for the BA.2 and BA.4 variants. The most stable RBD conformation is found for the BA.2 variant, which maintains the same behavior throughout most of the 1.5 µs simulation. This behavior was similar for all replicas, as shown with the WT in the presence or in the absence of glycans (Appendix A). Glycans induced a faster deviation from the initial starting structure, suggesting a moderate influence of glycosylation on the dynamic stability of the complex. An increase in RMSD values was observed in some replicas, which was induced by the motion of the c-terminal end of the RBD domain during the simulation (Appendix A). However, this part is not involved in the binding to hACE2 and should not be considered here, as it is linked to the S1/S2 cleavage site (or S2 domain) in the full-length protein. These high-level observations are related to the global behavior of the spike RBD in complex with hACE2 but are far from sufficient to explain the difference in infectivity observed between the variants. Thus, more detailed analyses of the MD simulations, at the atomic level or pair-wise interactions, have been conducted in order to refine our understanding of the differences between Omicron and Delta variants during the binding of RBD to hACE2.

### 3.2. Pair Interactions Study

The first series of PairInt calculations deals with one group of atoms made of the entire modeled spike RBD (residues 333-526) and the other group made of the nearly entire hACE2 receptor (residues 19-615). Results are shown in Figure 3 (black curves). The fluctuations in the potential energy values over time are strictly identical to what the MD engine actually handled during the simulation (PairInt parameters used here match the ones from the MD simulations). Therefore, one may be tempted to overanalyze such results and draw conclusions regarding the difference in binding strength between spike variants. However, this would not be correct. As explained in the Methods section, the energy measured by PairInt calculation corresponds to the enthalpic part of the interaction and should not be confused with the free energy of binding, which fundamentally cannot be evaluated directly on such a frame-by-frame basis. Actually, the energetics of salt bridges are expected to largely dominate PairInt values, and this should be kept in mind when interpreting differences in PairInt energies.

As shown in Figure 3, the non-bonded RBD-hACE2 interactions fluctuate between −200 and −250 kcal/mol for the WT, Delta and almost all Omicron variants, except for the BA.4 variant, which rather fluctuates around and above −200 kcal/mol. This suggests that binding with this particular variant relies much less on polar interactions. Two secondary observations can be made. First, the energetic fluctuations are lower in the Omicron BA.1 and BA.2 cases (the only Omicron variants bearing the Q493R mutation). This suggests that salt bridges formed with these two variants are more stable (but not necessarily stronger), either intrinsically (more favorable salt bridges between potentially different charged amino acids) or because they are supplemented by other stabilizing interactions. Second, for the BA.1 type, there is most certainly a large conformational change to be observed at t~550 ns, accompanied by an elevation in the energetic level and, thus, likely corresponding to the destabilization of the global electrostatics network. This hypothesis needs to be confirmed by further observations obtained through the other analyses performed in this study. Such a transition may also exist in the other complexes without being detected by the PairInt analysis, especially if it does not involve changes in polar interactions or if it is masked by larger global energetic fluctuations.

#### 3.2.1. Focus on Q/K/R Spike Residue at Position 493

Q493R is one of the spike key mutations (with S371P, S375F, Q498R and Y505H), not observed in previous strains, and that defined the Omicron family when it emerged. Most interestingly, this mutation had been identified by lab experiments designed to find which mutations would confer a stronger spike affinity to human hACE2 [44]. Furthermore, another synthetic variant (MU10) used to design a murine model better matching SARS-CoV-2 behavior in humans [75] had the Q493K mutation instead (also Q499T and P500T). This mutation was erroneously mentioned as part of Omicron BA.1 in initial WHO alerts, which led us to design an incorrect model (BA.1^+Q493K^). Nevertheless, the behavior of this model appeared particularly interesting later. Indeed, the later Omicron variants BA.4 and BA.5 (as well as more recent BA.2 sub-variants such as BA.2.75) feature a R493Q reversion, demonstrating that this position, 493, is of particular interest. From the PairInt energy curves (Figure 3, colored curves), the interaction between residue 493 and the entire hACE2 followed the same profile between WT, Delta and BA.4 (which have a Q at position 493) with values fluctuating around −30 kcal/mol, indicating persistent non-bonded interactions, most likely hydrogen bonds. From 250 ns, the energy was more constant and stable for the BA.4 variant compared to the other Q493-bearing variants. In the case of BA.1 and BA.2 variants (Q493R mutation), the interaction energy was stronger than in WT, Delta or BA.4 with values below −100 kcal/mol, suggesting the formation of a salt bridge. Moreover, it should be noted that fluctuations were observed after 750 ns for BA.1, leading to a stronger interaction energy, while this event was not observed for BA.2. This variant displayed a slightly weaker interaction energy (centered around 55 kcal/mol). No major difference could be detected between Delta and BA.4. Interestingly, the transient BA.1^+Q493K^ showed two energy states (−130 kcal/mol and −50 kcal/mol), with huge variations throughout the simulation. These variations reflect the formation and breaking of a salt bridge, especially after 750 ns and 1250 ns, that corroborates the conformational transitions observed in the 2D-RMSD map (Figure 2).

Focusing on the BA.1^+Q493K^ S/hACE2 interaction fluctuations, shown in parallel with the K493 contribution (Figure 3, red curve), clearly shows that the BA.1^+Q493K^ should be considered an outlier compared to other systems. Indeed, the K493 energetics strongly dominate the entire spike binding to hACE2, to the point that the instability of this particular interaction (entropic effects of solvent exposure may be at work) is directly transferred to the entire assembly. If the K493 salt bridge were to break (which is not seen in the MD trajectory), the entire S-hACE2 association would be expected to disintegrate very rapidly. The fact that the Q493K mutation has never been seen in any dominant SARS-CoV-2 strain in the wild to date (2024-09) may be due to the fact that this mutation provides too much local strengthening of the S-hACE2 association, which is detrimental to the stability of the entire S-hACE2 interface, as seen here, and may also inhibit further dynamic processes necessary for viral fusion.

#### 3.2.2. Residue–Residue PairInt Calculations

The key components of the non-bonded interaction energy were further characterized by PairInt calculations at the residue level to better highlight the most contributing residues either from hACE2 or RBD. In these representations (Figure 4), each row labeled with one hACE2 residue corresponds to the energy computation of the interaction between this residue as the first group of atoms, with the complete spike RBD (residues 333-526) as the second group of atoms, and vice versa. Protein regions showing very low interaction energy were regrouped in areas such as A.I, representing the interaction energy of all residues included within this region. Figure 4 provides a simplified overview of PairInt results from the point of view of hACE2 (residues 19-615) on the one hand and of spike RBD (residues 333-526) on the other hand for the WT, Delta and Omicron family variants.

Complete interaction profiles are shown in Appendix A. PairInt calculations were performed in triplicate for the WT to rule out any potential bias coming from the initial structure used for simulations and resulting in quasi-identical profiles (PairInt calculations of three additional replicas on top of the original simulation are shown in Appendix A). As described above, the presence of a few glycans may affect the non-bonded energy with additional polar interactions and could change the PairInt profiles. For this purpose, three more replicas were performed (duration of 1.5 or 2 µs) with subsequent PairInt calculations computed for the RBD-hACE2 WT complex including glycans and showing highly similar interaction energy profiles (as shown in Appendix A).

Interestingly, a secondary mode of interaction already suggested in Figure 3 for the BA.1 system, in which the contribution of R493 to the overall interaction potential is enhanced but to the detriment of the overall spike interaction with hACE2, is seen in Figure 4, resulting from the breakage of the R498 salt bridge with hACE2 residue D38, leaving R493 to form a ‘double salt bridge’ with both E35 and D38. This is not dissimilar to what is observed with K493 in the BA.1^+Q493K^ simulation, although to a lesser extent energetically.

#### 3.2.3. Identification of hACE2 Hot Spots for Spike Binding

In hACE2 (left column on each panel of Figure 4), five distinct areas are targeted by the different spike variants investigated here, corresponding to well-defined secondary structure elements. (1) The primary interaction area targeted by spike RBD is region A.I (residues 19-52), the hACE2 N-terminal helix, which involves seven charged residues (E23, D30, K31, H34, E35, E37 and D38) that can form salt bridges with spike residues. These seven residues are not involved all at once, and the different spike variants promote their involvement in distinct combinations. For example, strong signals are observed for E23 and D30 in the WT trajectory and for D30 and K31 in the Delta variant. For Omicron BA.1 and BA.1^+Q493K^, the strong signals are observed for E35 and D38, whereas for the BA.2 and BA.4 variants, the strong interactions are for E35 alone and D38 alone, respectively. It must be highlighted that among the nine polar residues in this N-terminal helix, two of them (E22 and K26) do not appear to be targeted by any spike variant (see details in Appendix A). (2) A secondary polar binding area named A.II (residues 347-359) is formed by a short loop between two small beta strands, 347-352 and 355-359. It is centered on two neighbor residues of opposite charge: K353 and D355. K353 is surrounded by two glycine residues, providing significant flexibility to the GKGDF short sequence. Figure 4 shows that most spike RBD variants are able to exploit this particular hotspot on hACE2 for binding. (3) Three other areas (A.III.1, A.III.2 and AIII.3) are considered in Figure 4, although less important from the perspective provided by our PairInt analysis, as no charged residue is directly involved. Yet, signals corresponding to stable interactions are being consistently monitored in all studied complexes. In the A.III.1 segment (residues 79-84) of the second hACE2 helix (residues 72-88), the most important signal is observed for residue Y83. In the A.III.2 loop (residues 324-330), the most important signal is observed for E329 (Appendix A) but is too weak to correspond to a salt bridge. In the mostly helical region A.III.3 (residues 386-396), the most important signals are observed for the BA.2 and BA.4 variants.

#### 3.2.4. Identification of Distinct Regions of Spike RBD Involved in the Binding to hACE2

The right column of all panels in Figure 4 shows the results of PairInt analysis on various residues and regions of the spike RBD interacting with hACE2 (residues 19-615). In fact, the spike RBD non-bonded interactions involve a continuous sequence, roughly defined as the 400–510 region, as segments 333–399 and 511–526 from the spike RBD are positioned too far away from the hACE2 monomer to provide discernible signals of non-bonded potential. The PairInt results show that the spike key residues greatly vary between variants. The residues and regions analyzed in Figure 4 are one isolated key residue (K417), one major interacting area (S.I) and two secondary binding hot spots (S.II.1 and S.II.2). (1) The K417 residue can form highly stable salt bridges with hACE2 in the WT spike and in the Delta variant, but this interaction is annihilated by the Omicron-specific K417N mutation. (2) The S.1 region (484–505) is clearly the major binding region of all spike variants compared with the WT spike, except for BA.1^+Q493K^ and BA.4, in which the signal is similar to WT. This area concentrates many mutable spike residues, with five of them of special interest, since they form significant electrostatic interactions with hACE2 in at least one spike variant (positions 484, 493, 498, 501 and 505). (3) A secondary highly mutable area (since Omicron) is the 438–458 sequence (S.II.1), where the strong signal observed for residue K458 in the WT spike and to a lesser extent in the Delta variant decreases significantly with Omicron variants to the profit of stronger signals for residue Y449 in the Delta variant and for both Y449 and K440 residues in the Omicron variants. (4) The other secondary interaction area (S.II.2) is the 473–478 short loop, in which a much stronger signal can be detected for residue N477 in the Omicron variants when compared to residue S477 in the Delta variant and WT spike.

### 3.3. Hydrophobic Interactions

Hydrophobic interaction patterns are detailed in Appendix A for spike RBD (panel A) and hACE2 (panel B) residues, analyzed separately along the six MD trajectories considered in this study. A graphical synthetic summary is represented in Figure 5, where each bar represents, for a given residue, the number of frames in which this hydrophobic contact has been observed. The upper panel is for the spike RBD and the lower for the hACE2 component. A total of 11 residues in spike RBD and 10 in hACE2 display counts of frames greater than 50% of the trajectory. Interestingly, for the spike RBD component (Figure 5 upper panel), five residues (F456, Y473, A475, F486V and Y489) display hydrophobic interactions in more than 12,000 frames (80%) in all six simulations. For Omicron variants, three additional mutated residues (Q498R, N501Y and Y505H) are detected above this threshold of 80% of the frames, in agreement with the corresponding mutations (one exception for Y505H in BA.2). It is worth noting that for the non-mutated Q498 and N501, the counts of frames are very low (below 2000) in WT and in the Delta variant, whereas for the non-mutated Y505, the counts of frames are slightly lower than the threshold of 80%. Finally, the Q493 position was found in more than half of the frames, except for BA.1^+Q493K^ and BA.4 showing a lower contribution at that position. For the hACE2 component (Figure 5 lower panel), four residues (T27, F28, K31, L79) display hydrophobic interactions in more than 12,000 frames in all six simulations, and two (M82, Y83) are found in more than 12,000 frames in all simulations except for the BA.4 variant, for which a reduction in the contact frequency is observed. It must be noted that the main regions identified by the PairInt analysis also contain hydrophobic residues, which contribute to the spike RBD binding to hACE2. For instance, the S.I region encompasses F486 and Y489 residues in all variants and the three additional contacts induced by the mutations in Omicron (Q498R, N501Y and Y505H). As for the two secondary binding hot spots, the three residues F456, Y473, A475 were found to contribute to the formation of hydrophobic contacts. The same observation can be drawn for their counterparts in the hACE2 side, with the main hot spot ranging from E23 to D38, including residue K31 (both forming electrostatic interactions and hydrophobic contacts) and T27 or F28 also belonging to this region. As for residues M82 and Y83 involved in more than 80% of the frames, these positions were also identified as a hot spot in the PairInt analysis (Figure 4 and better seen in Appendix A), meaning that these positions play an important role in the binding.

### 3.4. Binding Free Energies Evaluation

To gain complementary information about the binding affinity of RBD to hACE2, we compute the binding free energy using the MM/GBSA method for the ancestral and variant strains (Figure 6A). The ΔG values were calculated in triplicate by selecting about 100–200 ns extracted from the last part of each simulation. The binding affinity for WT (−20.9 ± 5.1 kcal/mol) was found to be weaker than that obtained with the Delta variant (−24.3 ± 6.1 kcal/mol), and these affinities were stronger for all Omicron variants, especially for BA.2 (−33.4 ± 8 kcal/mol) or BA.4 (−33.8 ± 5.2 kcal/mol), indicating a more robust and sustainable interaction between both partners. These results corroborate previously published data, showing, in general, a stronger affinity for Omicron RBD than WT [34,36,76]. The per-residue energy decomposition (Figure 6B) did not reveal major differences between original and variant strains, except for a few subtle changes observed for S477N (BA.2) or Q493R (BA.1) and N501Y mutations. The most remarkable variations (from Delta to Omicron) were located at positions 501 and 505, with larger energetic contributions of these two mutated residues in Omicron strains. The strong contribution of Y501 is in good agreement with the data obtained by hydrophobic interaction measurements. Surprisingly, the energetic contribution of R493 was not very high except for the BA.1 variant in comparison to WT or Delta strains, whereas residue R498 was more involved in the binding energy for BA.1 and BA.4 (Figure 6B).

### 3.5. Contact Principal Component Analysis

We performed cPCA using the contact data ensembles from our six different simulations over the last 700 nanoseconds (from 800 ns to 1500 ns according to the stability of PCs values; see the Methods section for details). First, to select the most appropriate number of components, we report the evolution of the explained variance ratio of the first ten components (Appendix A). The first five components particularly stand out, each of them containing from ~9 to ~5% of the explained variance ratio (cumulative total ~35%), while all subsequent components contain less than 2%. The relatively low proportion of explained variance observed in the cPCA can be attributed to two significant factors. Firstly, the problem is highly dimensional, with the potential for contacts to span up to n_atoms_^2^, greatly surpassing the dimensionality of more conventional PCA, such as Cartesian coordinates (n_atoms_) or dihedral angles (4n_atoms_). Moreover, the localized nature of contacts makes them highly sensitive to thermal fluctuations. It should be noted that here, one of the aims of PCA is to eliminate noise from the data, which allows for concentrating on reproducible motions. This suggests that the essential information about the dynamics is found inside the first five principal components, in total accounting for ~35% of the system variance. In Appendix A, we report the time evolution of the PC1-6 values for each simulation. For each simulation and PC, the time evolution of the PC values is mostly stable within the last 700 ns, which suggests that the MD simulations have appropriately converged and are suitable for cPCA. This is confirmed by the calculation of PC values during the complete simulations, as compared to the last 700 ns (compare Figure 7A,B).

To better understand the information captured within these six principal components, we represent, in Figure 7, the biplots of the simulations projected to the PC1-PC2 (panels A, B), PC3-PC4 (panel C) or PC5-PC6 (panel D) space. Interestingly, the first principal component for the WT and Delta variants has a strong positive PC1 value (between +50 and +150, panel B), while all Omicron variants have either negative values (for both BA.1 variant, between −50 and −150) or values close to zero (BA.2 and BA.4, between −25 and 25). This suggests that mutations encountered in Omicron BA.1 have the greatest impact on system contacts compared to mutations found in BA.2 and BA.4 variants.

Moreover, when representing the PC1 network (Figure 8A), most of the contacts of interest that are located near the interface are directly correlated with these mutations. In a given PC, frames are separated and categorized between negative and positive values. The coefficients of the corresponding eigenvector each register one contact, and their sign directly indicates if the contact is stronger in the frames with a positive PC value than in frames with a negative PC value. Therefore, the absolute value of this coefficient indicates how pronounced the overall effect is. Here, starting with the S477N mutation present in all Omicron variants, the N477 residue in the RBD shows a stronger contact with residue S19 from hACE2 compared to non-mutated S477 present in the WT and Delta strains (Figure 8A).

This suggests that this mutation directly increases the binding of spike RBD with hACE2. In addition, due to the Q493R/K mutation, present as Q493R in BA.1 and BA.2 and Q493K in BA.1^+Q493K^, the interaction is strengthened in all Omicron variants with residues E35 and D38. Therefore, the mutation of glutamine, a neutral residue, into either an arginine or a lysine (both positively charged) reinforces the interaction between residue R/K493 and E35 and D38 in a double salt bridge. The slight difference in PC1 values between the two BA.1 strains (around −75 for BA.1 and −100 for BA.1^+Q493K^) can be explained solely by this difference of mutation. This also suggests that the Q493K mutation is more stabilizing than the Q493R mutation. As shown in Figure 8, within the RBD, the S375F mutation, found in all Omicron variants, induces a stronger F375-Y508 contact in Omicron variants than the S375-Y508 contact existing in the WT and Delta variants (Figure 8A). This suggests that the S375F mutation induces this interaction through a π–π contact. Interestingly, Y508 is located near another mutation spot in the RBD, notably G496S, Q498R, N501Y and Y505H, which are parts of a dense contact area in the RBD. The main changes at the interface are enhanced by the transition from N501-K353 contact in the WT and Delta to Y501-K353 in Omicron strains (Figure 8A). This contact was demonstrated to be the primary change in Alpha, Beta and Gamma variants [39].

For PC2 (Figure 7B, *y*-axis), Delta variant and Omicron BA.2 and BA.4 strains display mostly negative values (PC2 value between −25 and −100). In contrast, the WT displays highly positive values (PC2 value between 125 and 175), while the two BA.1 strains have slightly positive values (between 0 and 50). In PC2N (Figure 8B), only three contacts are found at the interface: G476-T20 (typically stronger in the WT than in Delta and BA.2), Q498-Y41 and T500-D355 (typically stronger in Delta and BA.2/4 than in the WT). A large contact patch is found within the RBD, very similar to our previous results concerning the Alpha, Beta, Gamma and Delta variants [39]. It is worth noting that the effect observed in BA.2/4 is not entirely replicated in the BA.1 variants. This dissimilarity between the two could explain why the latter was more prevalent than the former in the European Union.

In PC3 (Figure 7C, *x*-axis), the WT, BA.1^+Q493K^, BA.2 systems exhibit negative values (respectively, around −45, −55 and −80), while BA.4 shows slightly positive values (around 10) and Delta and BA.1 have stronger positive values (respectively, 50 and 100). Surprisingly, despite a single mutation difference between BA.1 and BA.1^+Q493K^, these variants show a strong opposition in this component. Intriguingly, residue Q493 is not involved in the PC3 network (Figure 8C). Still, near the mutation, there is a cluster of contact changes between the β5, β6 sheets and the β5-β6 turn. The proximity between this patch of contact changes and the Q493K mutation suggests an indirect effect of the mutation, leading to a different reorganization of the RBD. Interestingly, this cluster is also connected to the L452R residue (found in the β5 strand), which is mutated only in the Delta variant. Only two contacts at the interface show some disruption: Y505-K353 (stronger in Delta and BA.1 variants than in WT, BA.1^+Q493K^ and BA.2) and Q498-Y41 (stronger in the WT, BA.1^+Q493K^ and BA.2 than in the Delta and BA.1 variants).

In PC4 (Figure 7C, *y*-axis), the Delta and BA.1^+Q493K^ variants both have strong negative values (respectively, around −75 and −50), while BA.1, BA.2, WT show values close to zero (between −10 and 40) and BA.4 has a strong positive value (above 100). This component shows both the internal rearrangements of the spike RBD in the α2 and α4 helices and a small disruption at the interface between residue F/V486 in the spike RBD and residues M82 and Y83. This shows that mutation F486V, which is exclusive to the BA.4/5 strains, reduces contacts at the interface without disrupting the surrounding interface too much. Interestingly, this residue is key in the cross-talk between mutations L452R and T478K in the Delta variant, while mutation L452R is also present exclusively in the BA.4/5 strains. Here, in contrast, the results suggest that the cross-talk between T478K and L452R is closed in the case of BA.4/5 strains and that the L452R mutation plays an additional role.

In PC5 (Figure 7D, *x*-axis), the PC values taken by most simulations have a stronger amplitude than in the latter components. This suggests that a part of the variance is explained by variation between simulations, while another part of this variability is explained by motions occurring within the simulations. This, added to the fact that components are of decreasing importance in the explained variance, makes it harder to interpret the PC network shown in Figure 8D. In fact, the next component, PC6 (Figure 7D, *y*-axis), shows strong oscillations in the WT (between −150 and 150), while other simulations only oscillate between −30 and 30. Thus, the sixth component is representative of a dynamic motion occurring principally in the WT. No PCN was derived from this component.

### 3.6. Cross-Correlation Analysis

As a final analysis to better highlight the role of each variant on the overall flexibility and dynamics of the hACE2-RBD complex, we performed normalized linear mutual information (nLMI) analysis on all the generated trajectories, as depicted in Figure 9A. The reported values of nLMI are color coded from dark blue (no correlation) to dark red (maximum correlation). Table 2 (derived from Appendix A) shows a comparison of the average nLMI values for the residues within RBD, within hACE2 and between RBD and hACE2, and between the WT system and the five variants considered in this study (Delta, BA.1^+Q493K^, BA.1, BA.2, and BA.4). This comparison highlights specific patterns that could divide the variants into three groups: (i) Delta, (ii) BA.1^+Q493K^ and BA.1, (iii) BA.2 and BA.4. For the Delta variant, while we observed a significant increase in correlations within the RBD (by 0.07), decreases in correlations were observed within hACE2 (by 0.06) and between the two proteins (by 0.01) in comparison to the WT strain, with average nLMIs of 0.64, 0.59 and 0.55, respectively.

In the case of BA.1^+Q493K^ and BA.1, an overall decrease in average nLMI values was observed for RBD (both by 0.10) and between the two proteins (by 0.04 and 0.03) with respect to the Delta variant. The average values remained the same within the hACE2 for BA.1^+Q493K^ and increased by 0.01 for B1.1. Finally, both BA.2 and BA.4 showed a decrease in the average nLMI within the RBD (by 0.04 and 0.05), while the average values were increased within the hACE2 (by 0.06 and 0.04) and between RBD and hACE2 (by 0.04 and 0.02), as compared to the Delta variant. To better highlight the residues that promoted the highest correlations, we analyzed the eigenvector centrality within the residues of RBD and hACE2 for the WT and variants, as reported in Figure 9B,C. All the mutations within the RBD of Delta and Omicron are included in the plot as dotted and solid horizontal lines. The results suggested that residues with high eigenvector centrality are also sensitive to mutations. Moreover, we investigated the WT system to identify dynamically crucial residues. From the analysis of eigenvector centrality, we extracted the residues with values within the top 10% in both RBD and hACE2 and mapped them on the structure (see Figure 9D). In the RBD, the majority of them are placed at the interface with hACE2 and within the receptor binding motif (RBM). However, residues with high eigenvector centrality within the hACE2 are not located at the interface but create a cylinder-like region in the middle of the protein, suggesting a long-distance effect.

### 3.7. Visualization of the Main Contacts Identified by PairInt, MHP and cPCA Analyses

The main results of PairInt, hydrophobic contacts and cPCA analyses were mapped to the 3D structure of RBD/hACE2 complexes in order to illustrate the major key interactions. The structures were extracted from the MD simulation trajectories, and the main occurring contacts are indicated (Figure 10) in order to better understand the differences between Delta and Omicron variants. As detected by the PairInt analysis (Figure 4), the key electrostatic interactions for Delta were mediated by K417 with D30, while other contacts persisted between Q493 and K31/E35 or between Q498 and K353; this lysine also shares a bond with N501 (Figure 10A). Due to the K417N mutation in Omicron, this interaction was not detected anymore in Omicron variants, while the major interaction (E35-Q(R/K)-493) remained stable throughout the simulation. In BA.1, the N-terminal α-helix of hACE2 (residue S19) makes contact with N477, stabilizing this region (Figure 10B). R493 was able to share interactions with both E35 and D38 during the simulation (Figure 10C). In BA.2, R493 also shares this interaction in the beginning of the simulation with both E35 and D38 but only forms a steady interaction with E35. In BA.4, Q493 makes hydrogen bonds with both E35 and K31. In addition, the Q498R mutation reinforces the electrostatic contribution between R498 and D38, highly important for BA.4, in which the R498-D38 salt bridge was shown to be transient (Figure 10D).

From the analysis of hydrophobic or van der Waals contacts, the method revealed the presence of two hydrohphic clusters located at each edge of the central α-helix of hACE2, encompassing residues T27, F28, K31, Y41, L79, M82, Y83 for hACE2 residues. These residues were found to form stable contacts mainly with F456, V486, Y489, Y501 and H505 from spike RBD (Figure 10E). As already illustrated in Figure 8, the cPCA method allowed us to identify the previoulsly described interactions, especially at the point mutation S477N or Q493R/K or F486V with M82/Y83. Moreover, it also highlighted an internal connection within the spike RBD between F375 (S375F mutation present in all Omicron) and Y508 (Figure 10F). This intramolecular interaction may play a role in stabilizing the two adjacent β-sheets and could contribute to the global binding stability.

## 4. Discussion

In this study, we applied long-scale explicit-solvent MD simulations (total of 20.5 µs) of the S-hACE2 system for six different spike sequences (WT, Delta, Omicron BA.1, BA.1^+Q493K^, BA.2 and BA.4/BA.5). In order to better understand the structural and dynamical aspects of S RBD evolution, we used a combination of very distinct, but complementary, MD trajectory analysis approaches.

Assessing all those approaches, it appears that at least 500 ns of MD is required to obtain the S-hACE2 complex near a stable conformational basin, especially for BA.1 and BA.4. Furthermore, interesting configurational changes only spontaneously occur on the microsecond scale. This is significantly longer than what we expect to observe on systems of similar size, probably due to the spike RBD being intrinsically a shape-shifting molecular machine that requires serious simulation time, starting from a low-temperature stabilized state, as seen in the Protein Databank models.

Basic MD analyses (RMSD, PCA) indicate that the differences between spike variants as seen from MD simulations are more marked than what may be inferred from comparing experimentally derived models. Indeed, each system showed a distinct dynamic profile, with Omicron BA.1, BA.2 and BA.4 displaying narrower ones compared to WT and Delta.

It is unclear whether the strength of hACE2 binding increased or decreased with Omicron, except when computing the binding free energy with implicit solvation methods like MM/GBSA, showing a stronger affinity with Omicron compared to WT and Delta strains. However, electrostatic-focused interaction energy fluctuations from our 1.5 µs long MD trajectories suggest that there is no significant change between the systems in the S-hACE2 electrostatics globally, except maybe for a slight weakening with BA.4. This observation is directly attributable to the Q493R reversal, which is apparently compensated by other interactions (two H-bonds with K31 and E35 on hACE2). Electrostatics for the Omicron assemblies appear slightly more dynamically stable than those of WT and Delta. BA.1^+Q493K^ is a notable exception, with analyses depicting an unstable system and providing some clues as to why a lysine was not observed at position 493 in any competent SARS-CoV-2 strain. Even more surprisingly, the latest variant (KP.1/24C, not present in this study) carries the Q493E mutation, introducing a negative charge residue instead of a neutral one. This confirms that this position is highly variable, without resulting in a loss of binding affinity (E493 may still interact with K31 from hACE2, see Figure 10G).

We investigated S-hACE2 binding at the amino acid residue level using different, complementary approaches. The most direct one (PairInt, directly using MD potentials) is suitable for monitoring electrostatics and large conformational changes, which is particularly relevant for the S-hACE2 systems. In contrast, the MM/GBSA method was not as sensitive as PairInt for detecting lasting electrostatic interactions such as salt bridges (only K353 clearly identified) when analyzing per-residue energetic contributions. Nevertheless, these calculations allowed us to confirm important interactions at hot spot mutational positions of spike (F/V486, N/Y501 and Y/H505). The mutation N501Y has been identified since the alpha variant and is responsible for enhanced binding to ACE2 by inducing local conformational changes at the interface [77]. The key residues responsible for binding, on both S and hACE2, are already highlighted in experimentally derived models. Those describe how key molecular features of S-hACE2 evolve with S mutations, with some locally weakening RBD binding being compensated by others that may promote previously unseen pairings, thus leading to significant changes in the S-hACE2 binding mode [26]. Our simulations provide supplementary knowledge by allowing us to assess the stability and relative strength of the interactions, especially salt bridges. Indeed, regarding the region of prime interest for BA.1 key mutations Q493R and Q498R, two salt bridges, R493-E35 and R498-D38, are formed. Interestingly, the R493-E35 salt bridge is significantly more stable according to our analysis. A major secondary mode is also seen, where R493 is able to “steal” D38 from R498, forming a dual salt bridge, also observed with K493 from BA.1^+Q493K^, and very briefly, at the beginning of the BA.2 simulation. This new conformation, which is energetically and dynamically less favorable, cannot be guessed from the analysis of the reference static models. The BA.2 trajectory shows no stable salt bridge formed by R498, while the R493 one does not hold for the total duration of the simulation. This clearly diverges from the later-released structure, which depicts BA.2 binding to hACE2 similarly to BA.1 [42]. Therefore, three distinct MD simulations (counting BA.1^+Q493K^ depicting K493 as even less viable than Q493) suggest, in different ways, that the presence of a positively charged residue at position 493 on spike, in the process of forming a salt bridge, may also induce secondary unfavorable behavior, thus explaining why Omicron later reversed the Q493R mutation. The BA.4 simulation also suggested that Q493 may allow for a transient return of the R498-D38 BA.1 salt bridge, which is observed on the X-ray model released later [49].

We also identified two regions where S-hACE2 binding involves two stable hydrophobic contacts (key spike residues being F456, V486 and Y489; Y501 and H505) that are of prime interest upon comparing Delta to Omicron. The F486V mutation in the BA.4 strain also seems to prevent synergistic cross-talk between mutations L452R and T478K that was previously discovered in the Delta variant. Correlation analysis using normalized linear mutual information (nLMI) with WT as a reference classified the studied variants into three distinct groups, Delta, BA.1, and BA.2/4/5, the latter group being, surprisingly, the closest to WT. This suggests that the two S Delta mutations individually had a much larger impact than those on Omicron strains, confirming that the initial Omicron B.1.1.519 spike sequence was the result of a very different evolutionary process compared to pre-Omicron variants of concern, mutating by clusters, yet managing to maintain the global structure and binding efficiency to hACE2.

## 5. Conclusions

Since numerous publications investigating spike using MD simulations are available [78], a question one might ask regarding the present study could be “What does this bring compared to all the others?” We would stress that we used MD in order to investigate and finely compare the structural dynamics of spike variants when bound to hACE2, rather than trying to compute the corresponding energetics, which is often the main focus elsewhere. We exclusively aim to provide a more in-depth understanding of the structural mechanisms in play at the molecular level during spike evolution, MD being intended to complement existing knowledge from reference experimentally derived models.

Modern MD-based studies should use up-to-date models as starting points and not resort to dangerous methodological shortcuts. Unfortunately, this is rarely the case, as shown by the discrepancies listed in Appendix A [25,26,34,36,41,79,80,81,82,83,84,85,86,87,88,89], suggesting that experimental protocols also have some issues regarding the determination of binding affinity constants. Indeed, although the number of publications focusing on the difference between Omicron and Delta variants using theoretical approaches is quite abundant in the literature, their conclusions are sometimes contradictory. These discrepancies could also be explained by the different settings used in the simulations, such as the crystal structure used as the starting material, the duration of the simulation, the time step of the integration, the force field used, the non-bonded energy calculation (frequency, cutoff distance) and the type of water model used as the solvent. One should carefully consider the parameters used and keep in mind the limitations of such theoretical methods. Another consideration comes from the salt concentration used in MD, as the electrostatic interactions play a major role in the binding of both partners. Lower salt concentrations than physiological ones may increase the strength of these interactions and exaggerate the electrostatic contribution. Obviously, this mode of interaction, mostly involving the electrostatics, is not exclusive to the spike/hACE2 complex as it is found in many interactions between two proteins (viral and cellular [90]) or protein with nucleic acids [91] or between protein and lipid membranes [92]. This phenomenon is often encountered in many viral replication cycles (HIV-1, HTLV-1, Chikungunya virus, SARS-CoV-2…) [93].

However, is quantifying the strength of spike binding to hACE2 critical for evaluating the threatening level of a given SARS-CoV-2 variant? There has been a staggering accumulation of evidence for post-Omicron spike mutations, being driven by the accelerated acquisition of resistance against antibodies [15,19,94,95,96,97,98,99,100]. For each major new variant, there is hardly any consensus on how the affinity of spike for hACE2 can change; in any case, this became of secondary interest. Unless another Omicron-like earthquake in SARS-CoV-2 evolution happens, in which case the virus evolutionary trajectory may drastically change again, “How much may spike improve its binding to hACE2?” is currently much less relevant than “How, and how long, may spike retain enough affinity for hACE2 while continuing accumulating resistance?” Our study ends with the BA.4/5 variant, and it should be emphasized that the Omicron family has continued to evolve since that time. Interestingly, since XBB and subvariants, we can note that most recent variants have collected all the specific mutations defining the major pre-Omicron strains (encompassing all variants of concern from 2020 to 2021), with the brief exception of the Lambda F490S mutation (Appendix A).

We shall realistically assume, from our perspective as molecular modelers that spike, the defining feature of all coronaviruses possesses a large “conformational reservoir” that may be exploited to retain enough hACE2 affinity in the process of mutating to counter the increasing antibody pressure. In this context, by exploring the structural plasticity in S-hACE2 complexes beyond static models, MD simulations may indirectly provide original clues in the process of anticipating SARS-CoV-2 evolution. However, this approach takes time, not to mention the required computational cost. MD simulations based on available PDB models may provide complementary knowledge to the scientific community at a moment when a new variant of concern may have superseded the one being investigated. As briefly mentioned in the Introduction, the gap between experimental data and in silico models was rapidly filled by deep mutational scanning studies performed on the SARS-CoV-2 Wuhan strain [32] and more recently on Omicron variants [33]. Interestingly, the key residues that have been identified in the spike RBD/hACE2 interactions in our study were specifically detected as point mutations that promote deleterious or beneficial effects on spike binding to hACE2. These data confirmed our predictions, especially for the residues at positions 417, 493, 498 and 501, for which mutations alter the binding affinity. Surprisingly, the latest Omicron variant (KP.3) has the R493E mutation, which results in opposite charge residues at this position, but the interaction can still occur by interacting with K31 of the spike RBD instead of E35 or D38 in other Omicron variants (precursors like BA.1/BA.2 or JN.1, for instance).

## Figures and Tables

**Figure 1 biomolecules-15-00541-f001:**
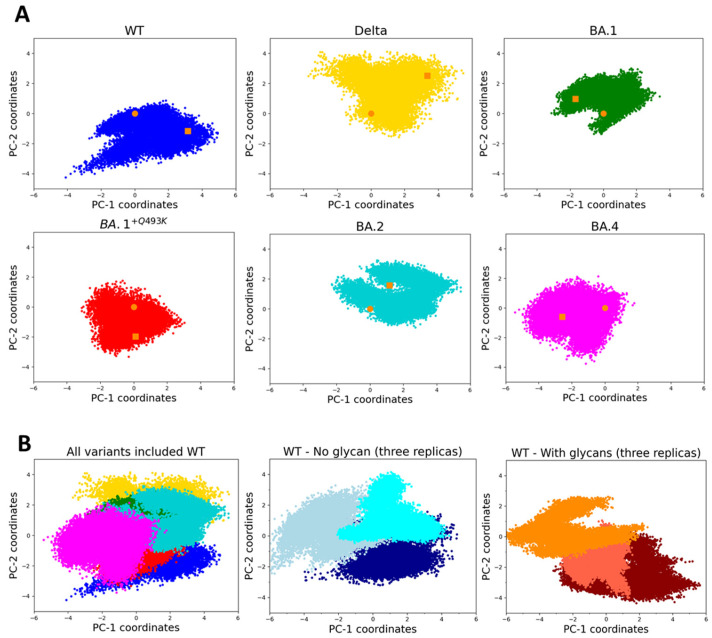
(**A**) PCA projections of the first two modes showing the conformational space visited by each individual RBD/hACE2 complex for WT (blue), Delta (yellow), BA.1 (green), BA.1^+Q493K^ (red), BA.2 (cyan) and BA.4 (magenta) variants (PCA computed on all backbone atoms using 15,000 frames or 1.5 µs, orange square and dot points indicate begin/end frame of the simulation, respectively). For all systems, the average variance explained by PC1 and PC2 is about 34% and 16% of the total variance, respectively (Appendix A). (**B**) PCA projection along the first two modes of all system (WT and variants, left panel), three replicas of WT (without glycan, middle panel) or three replicas of WT with glycans, right panel).

**Figure 2 biomolecules-15-00541-f002:**
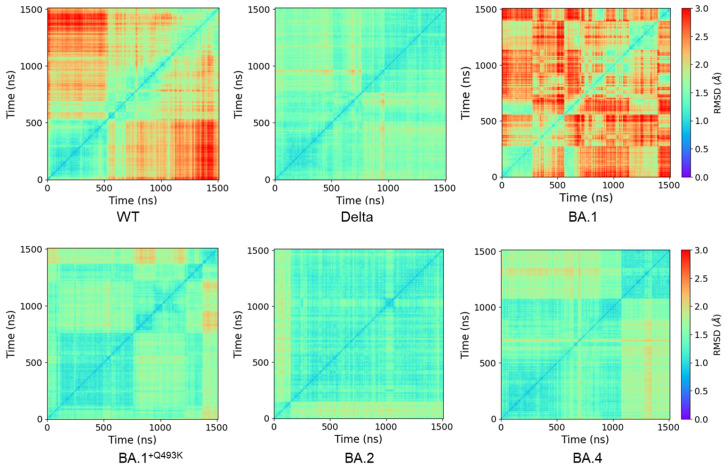
Two-dimensional RMSD matrix plots computed for the RBD partner for WT and variants highlighting the conformational transitions underwent by spike RBD during the simulation. RMSD were computed on all backbone atoms and using as reference frame the first frame of the trajectory (corresponding to the equilibrated structure).

**Figure 3 biomolecules-15-00541-f003:**
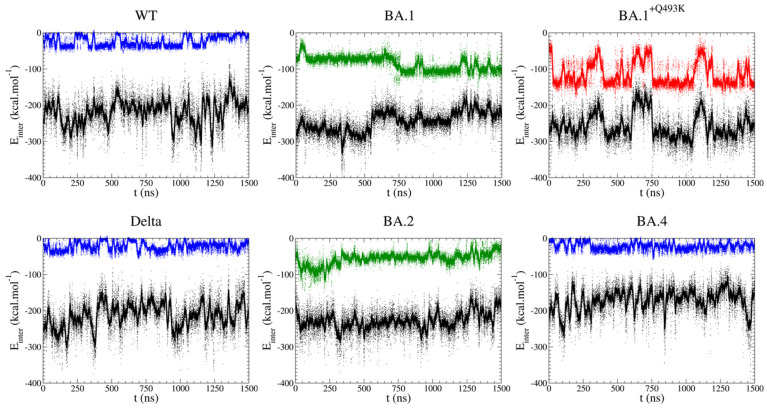
Non-bonded interactions computed from MD trajectories as described in the PairInt (PI) procedure. Potential total energies for spike variants (modeled residues: 333-526) and hACE2 (19-615) are shown as black lines. Each dot corresponds to one frame in the MD simulation, the curves being a running average over a 10 ns sliding window. In each plot, the top colored curve shows the interaction between the variable spike residue at position 493 (color-coded as follows: blue for Q, green for R and red for K) and residues from hACE2 (19-615).

**Figure 4 biomolecules-15-00541-f004:**
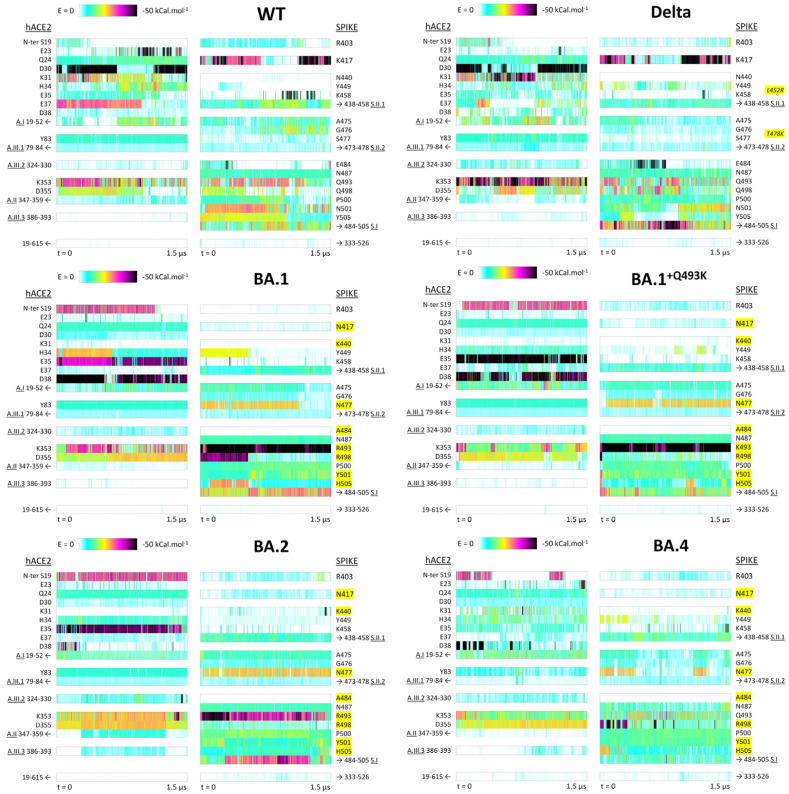
Non-bonded interaction energy decomposition by residue as a function of time during the MD simulations. Each row labeled with one hACE2 residue in the left column of each panel corresponds to the energy computation of the interaction between this residue as first pair member and the complete spike RBD domain as second group of atoms and vice versa for spike residues. The A.I, A.II, A.III.1, S.I, S.II.1 and S.II.2 interaction areas correspond to the cumulative residual part of these regions after subtracting the contribution of the above most-involved residues. Mutated spike residues from the original WT strain are highlighted in yellow.

**Figure 5 biomolecules-15-00541-f005:**
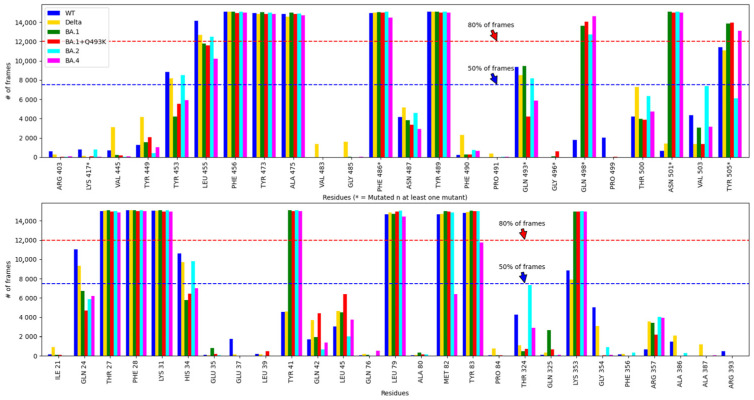
Evolution of hydrophobic interactions for WT and variants during the 1.5 µs simulation. Each bar is representative of the contact frequency of involved residues from either spike (**upper panel**) or hACE2 (**bottom panel**). Residues that are mutated between variants are indicated with a star.

**Figure 6 biomolecules-15-00541-f006:**
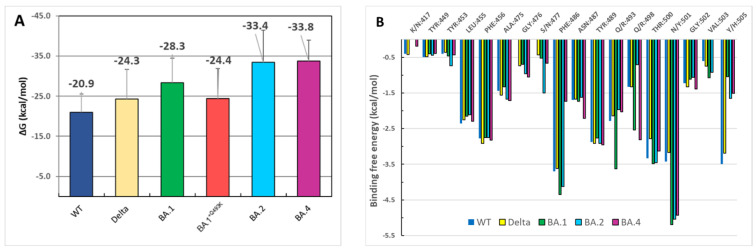
(**A**) Binding free energy values computed with MM/GBSA for the ancestral and variant strains (values are averages of three independent calculations on the same trajectory ± standard deviation). (**B**) Per-residue energy contributions computed from GB energy decomposition for WT, Delta/Omicron variants (BA.1/2/4).

**Figure 7 biomolecules-15-00541-f007:**
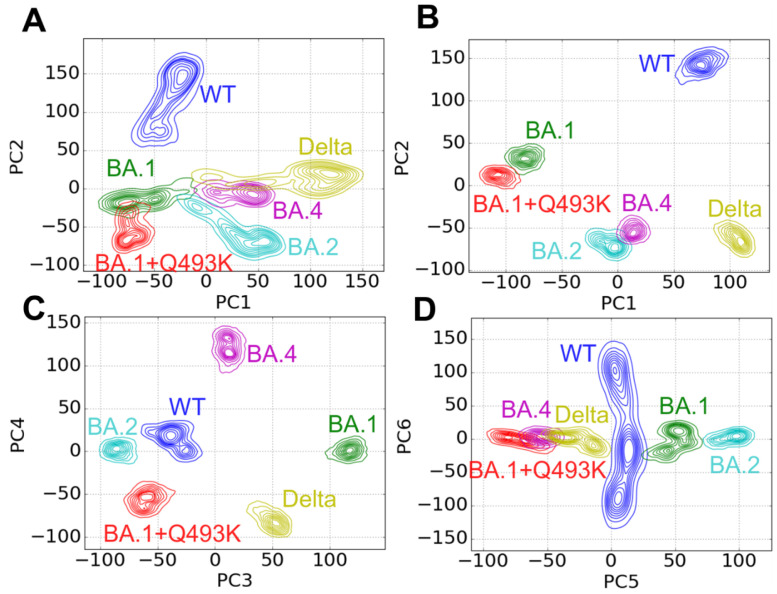
Contour plots showing equal distribution of MD frames (computed by kernel density estimate) within the cPCA coordinates space for WT (blue), Delta (dark yellow), Omicron BA.1 (green), BA.1^+Q493K^ (red), BA.2 (cyan) and BA.4 (purple) variants. (**A**) PC1 versus PC2 for the full-length simulations or (**B**) for the last 700 ns (see Methods for details), (**C**) PC3 versus PC4 and (**D**) PC5 versus PC6 for the last 700 ns.

**Figure 8 biomolecules-15-00541-f008:**
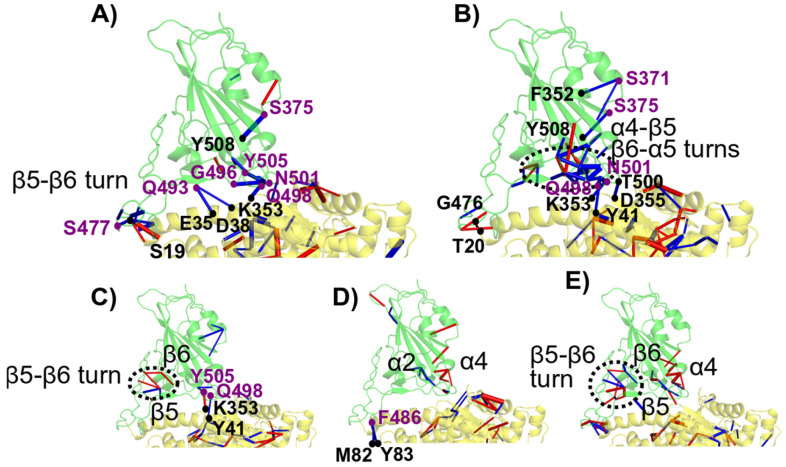
Contact networks corresponding to each eigenvector (cPCA) and represented on the 3D structure of the spike RBD (green) and hACE2 (yellow) components for (**A**) PC1, (**B**) PC2, (**C**) PC3, (**D**) PC4 and (**E**) PC5. Edge width is proportional to the influence of this contact in the eigenvector (using a 4% threshold of eigenvector contribution). Edges colored in red are representative of stronger contacts in variants with positive frames than in variants with negative frames, vice versa for edges colored in blue. All the variants cannot be compared in each PC as they depend on the signs of their frame distributions (as shown in Figure 7): (**A**) WT, Delta and BA.4 (positive) versus BA.1, BA.1^+Q493K^, BA.2 (negative). (**B**) WT, BA.1 and BA.1^+Q493K^ (positive) versus Delta, BA.2 and BA.4 (negative). (**C**) Delta, BA.1 and BA.4 (positive) versus WT, BA.2 and BA.1^+Q493K^ (negative). (**D**) WT, BA.1, BA.2 and BA.4 (positive) versus Delta and BA.1^+Q493K^ (negative). (**E**) WT, BA.1 and BA.2 (positive) versus Delta, BA.1^+Q493K^ and BA.4 (negative).

**Figure 9 biomolecules-15-00541-f009:**
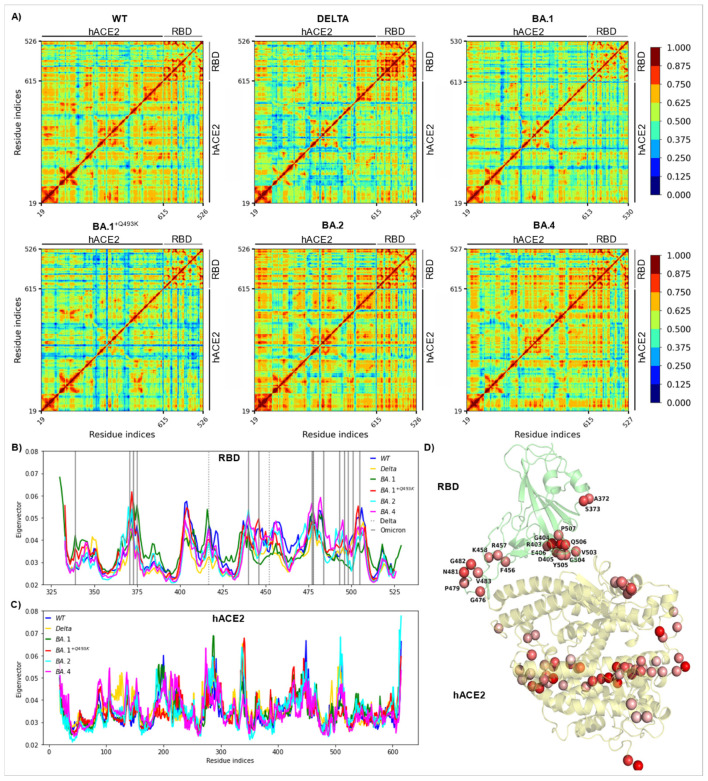
Correlation analysis of the WT and variant complexes of RBD-hACE2. (**A**) Normalized linear mutual information (nLMI) correlation plots. The nLMI plots are depicted for WT system, Delta and Omicron mutants (Delta, BA.1, BA.1^+Q493K^, BA.2, and BA.4). The correlation levels are shown with a color code from dark blue (no correlation) to dark red (high correlation). Projections of the eigenvector centrality on RBD (**B**) RBD and hACE2 (**C**) are depicted for all systems. The position of Omicron and Delta mutations are highlighted with solid and dotted gray vertical lines, respectively. (**D**) The eigenvector centrality values of the WT system are mapped on the structure and the residues representing highest values (top 10%) in RBD and hACE2 are shown with spheres.

**Figure 10 biomolecules-15-00541-f010:**
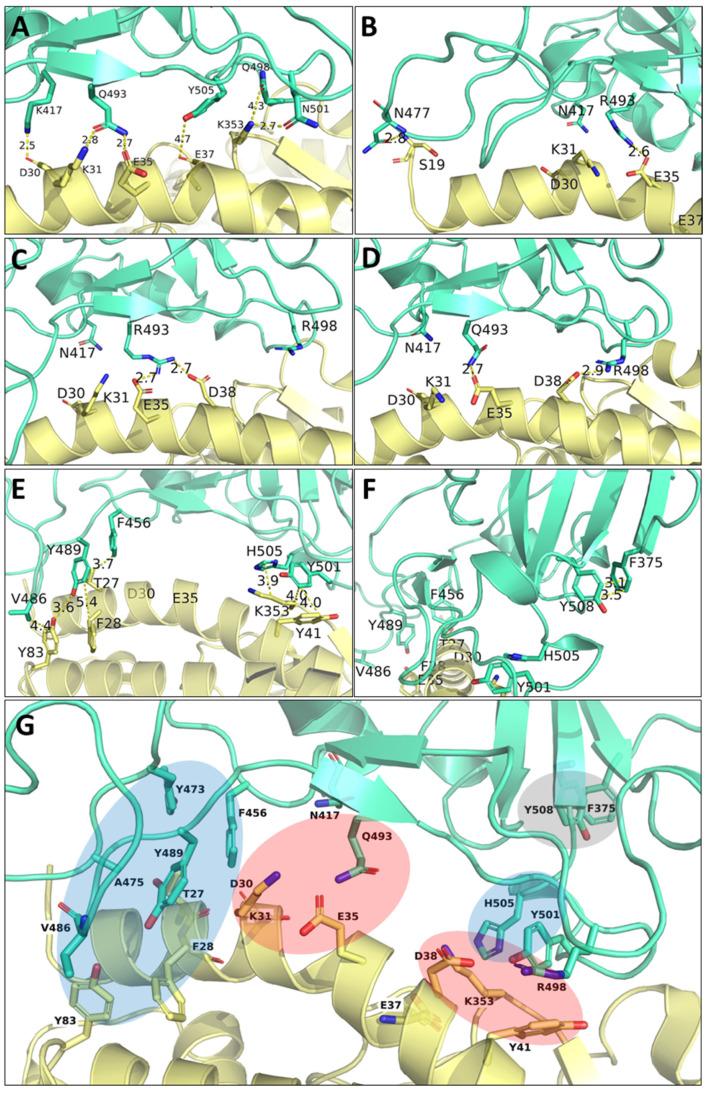
Illustration of the major contacts determined by the PairInt, hydrophobic contact and cPCA analyses. Structures were extracted from MD trajectories (spike RBD is depicted in green and hACE2 in yellow). Distances (Å) are purely indicative and should be considered as representative averages of these contacts since they can change during the simulation. (**A**) Main electrostatic interactions detected by PairInt highlighting the role of K417, Q493 and Q498 residues in Delta variant. (**B**) Impact of the mutation K417N in Omicron variants (absence of interaction with N417) and new contacts observed with N477-S19 and R493-E35. (**C**) Shared interactions detected for R493 with E35 and D38 in Omicron. (**D**) Impact of the Q498R mutation allowing a salt bridge between R498 and D38 in Omicron. (**E**) Major hydrophobic interactions identified by computing the MHP of each residue (with a cutoff distance of 4.2 Å applied for picking hydrophobic atom pairs) for Omicron showing the two hydrophobic patches forming by residues F456, V486, Y489 (RBD) with T27, F28, Y83 (hACE2) and Y501, H505 (RBD) and Y41, K353 from hACE2. (**F**) Zoom-in view of the intramolecular interaction within RBD between F375 and Y508 residues (S375F mutation occurring in all Omicron variants, contact identified by cPCA). (**G**) Summary of main interaction areas color-coded as follows: hydrophobic (blue), polar including salt bridges and hydrogen bonds (light red) and intramolecular bond (light grey) for the BA.4 variant.

**Table 1 biomolecules-15-00541-t001:** Available 3D structural models of RBD/hACE2 complexes solved by X-ray or cryo-EM at the starting time of our study for WT, Delta and Omicron strains. PDB IDs used for modeling variants in the current study are highlighted in bold.

Variants	PDB IDs	Reference	Method	Resolution (Å)
WT	**6M0J**	[41]	X-ray	2.5
	7KMB	[43]	Cryo-EM	3.4
	7BH9	[44]	Cryo-EM	2.9
	6LZG	[10]	X-ray	2.5
	7DQA	[45]	Cryo-EM	2.8
Delta	7W9I	[46]	Cryo-EM	3.4
	7V8B	(Yang et al., unpublished data)	Cryo-EM	3.5
	7WBQ	[25]	X-ray	3.3
Omicron	7WBP	[25]	X-ray	3.0
BA.1	7WBL	[25]	Cryo-EM	3.4
	**7T9L**	[26]	Cryo-EM	2.7
	7WK6	[47]	Cryo-EM	3.7
BA.2	7XB0 7XAZ	[48]	X-ray X-ray	2.9 3.0
	**7ZF7**	[42]	X-ray	3.46
BA.4	7XNQ 7XWA	[19] [49]	Cryo-EM X-ray	3.52 3.36

**Table 2 biomolecules-15-00541-t002:** The average nLMI and standard deviation values for all the studied systems. The corresponding heatmaps are displayed in Figure 9A. The nLMI values are averaged within RBD, within hACE2, between RBD and hACE2 complex, and over the complete map.

MD Simulation	Within RBD	Within hACE2	Between RBD and hACE2	Total
WT	0.64 ± 0.14	0.59 ± 0.12	0.55 ± 0.12	0.58 ± 0.12
Delta	0.71 ± 0.13	0.53 ± 0.13	0.54 ± 0.11	0.54 ± 0.13
BA.1^+Q493R^	0.61 ± 0.15	0.53 ± 0.13	0.50 ± 0.12	0.52 ± 0.13
BA.1	0.61 ± 0.14	0.54 ± 0.13	0.51 ± 0.10	0.53 ± 0.12
BA.2	0.67 ± 0.13	0.59 ± 0.13	0.58 ± 0.12	0.59 ± 0.13
BA.4	0.66 ± 0.15	0.57 ± 0.12	0.56 ± 0.12	0.57 ± 0.13

## Data Availability

Molecular Dynamics simulations were performed with NAMD v3.0 α9 (available from the Theoretical and Computational Group or https://www.ks.uiuc.edu). All protein structure files and simulation trajectories are available for download at https://doi.org/10.5281/zenodo.10144459.

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
