# Peer review of "Subtle Changes at the RBD/hACE2 Interface During SARS-CoV-2 Variant Evolution: A Molecular Dynamics Study"

_biomolecules, 2025, doi:10.3390/biom15040541_

Round 1
Reviewer 1 Report
Comments and Suggestions for Authors
The authors investigated subtle changes at the RBD/hACE2 interface during SARS-CoV-2 variant evolution, with a particular focus on Omicron variants and their comparison with earlier variants and the WT. Utilizing computational approaches, including molecular dynamics simulations, they conducted a comprehensive analysis of protein-protein interactions (PPI) to evaluate the impact of mutations on binding stability and interaction networks. This detailed and thorough study provides valuable insights into the binding feature of Omicron, contributing to a deeper understanding of SARS-CoV-2 evolution. Here are my comments.
1. In molecular dynamics simulations, the type and concentration of added ions directly influence the system’s charge balance, electrostatic interactions, and the stability of key salt bridges, making them crucial for computational accuracy. A common practice in molecular dynamics (MD) simulations is to add NaCl or KCl to mimic physiological conditions and ensure system neutrality. However, this study only added Na⁺ ions. What was the reason for using only Na⁺ ions instead of a full NaCl or KCl ionic environment?
2. This study provides exploration of SARS-Cov-2 variant evolution at the RBD/hACE2 binding interface, it is recommended to further discuss the role of electrostatic interactions in the discussion section. Previous studies have demonstrated that charge characteristics play a critical role in both nucleocapsid proteins of SARS-CoV and SARS-CoV-2 as well as in the interaction between the spike protein and ACE2 (e.g., Electrostatic features for nucleocapsid proteins of SARS-CoV and SARS-CoV-2; Revealing the mechanism of SARS-CoV-2 spike protein binding with ACE2; Computational biophysical characterization of the SARS-CoV-2 spike protein binding with the ACE2 receptor and implications for infectivity). Incorporating these findings could provide a more systematic understanding of how mutations influence RBD-hACE2 binding through electrostatic regulation, thereby enhancing the interpretation of Omicron’s evolutionary advantages.
3. There appears to be a potential inconsistency regarding the MD trajectory selection for different analyses. In Section 2.1, the authors state that MD trajectories were analyzed from 0 to 1.5 μs, with contact Principal Component Analysis (cPCA) specifically performed from 800 ns to 1.5 μs based on the stability of PC values. However, in Section 2.4, MM/GBSA calculations were conducted on 100-200 ns segments, without explicitly stating the criteria for segment selection. To ensure methodological clarity, the authors should ensure consistency across analyses.
4. The sentence “we performed MM/GBSA calculations in triplicate on segments of 100-200 ns of the simulation (in the range from 800 ns to 1500 ns) with the implicit solvation model” is ambiguous.
5. I do not fully understand the meaning of “blue: Q; green: R; and red: K” in Figure 3.
6. Minor Issues:
Please pay attention to small grammatical errors, such as “Total energy contributions per residues computed.
Author Response
The authors investigated subtle changes at the RBD/hACE2 interface during SARS-CoV-2 variant evolution, with a particular focus on Omicron variants and their comparison with earlier variants and the WT. Utilizing computational approaches, including molecular dynamics simulations, they conducted a comprehensive analysis of protein-protein interactions (PPI) to evaluate the impact of mutations on binding stability and interaction networks. This detailed and thorough study provides valuable insights into the binding feature of Omicron, contributing to a deeper understanding of SARS-CoV-2 evolution. Here are my comments.
We thank the reviewer for this general comment that highlights the quality of our study.
- In molecular dynamics simulations, the type and concentration of added ions directly influence the system’s charge balance, electrostatic interactions, and the stability of key salt bridges, making them crucial for computational accuracy. A common practice in molecular dynamics (MD) simulations is to add NaCl or KCl to mimic physiological conditions and ensure system neutrality. However, this study only added Na⁺ ions. What was the reason for using only Na⁺ ions instead of a full NaCl or KCl ionic environment?
Indeed, the salt concentration is an important parameter in molecular dynamics simulations especially when many electrostatic interactions are involved. We selected a salt concentration of 0.154 M (NaCl) in order to be as relevant as possible in respect to physiological conditions. We obviously added both ions (Na+/Cl-) to reach this concentration and only the electro-neutrality of the system was achieved with additional Na+ ions. We have rephrased this sentence in the Materials and Methods section to clarify this point.
- This study provides exploration of SARS-Cov-2 variant evolution at the RBD/hACE2 binding interface, it is recommended to further discuss the role of electrostatic interactions in the discussion section. Previous studies have demonstrated that charge characteristics play a critical role in both nucleocapsid proteins of SARS-CoV and SARS-CoV-2 as well as in the interaction between the spike protein and ACE2 (e.g., Electrostatic features for nucleocapsid proteins of SARS-CoV and SARS-CoV-2; Revealing the mechanism of SARS-CoV-2 spike protein binding with ACE2; Computational biophysical characterization of the SARS-CoV-2 spike protein binding with the ACE2 receptor and implications for infectivity). Incorporating these findings could provide a more systematic understanding of how mutations influence RBD-hACE2 binding through electrostatic regulation, thereby enhancing the interpretation of Omicron’s evolutionary advantages. Discuss about general feature of electrostatic interactions occurring during viral infections by different viral protein such as capsid or Spike.
Although the interaction of nucleocapsid with host cell components takes place several steps after Spike binding to hACE2 and viral fusion, this phenomenon of electrostatic interactions is indeed not exclusive to Spike or RNA binding protein and can be considered as a general mechanism governing many protein/protein or protein/nucleic acids interactions involved at different stage of the viral replication cycle. We have now discussed this general feature in our conclusion.
- There appears to be a potential inconsistency regarding the MD trajectory selection for different analyses. In Section 2.1, the authors state that MD trajectories were analyzed from 0 to 1.5 μs, with contact Principal Component Analysis (cPCA) specifically performed from 800 ns to 1.5 μs based on the stability of PC values. However, in Section 2.4, MM/GBSA calculations were conducted on 100-200 ns segments, without explicitly stating the criteria for segment selection. To ensure methodological clarity, the authors should ensure consistency across analyses.
The time window is different between the analysis and this may look like as an inconsistency but this is not the case as it has been done on purpose. Indeed, for classical analysis we used the full-length simulation time (as for RMSD, cartesian coordinates PCA or PairInt) but for contact PCA, we specifically selected the 800-1500 ns range as we could confirm the convergence and stability of the PCs to ensure that the contact networks described by each PC have been accurately determined. In absence of stability of the PC values, the contact PCA networks will be inaccurate in amplitude.
As for MM/GBSA, the calculation of binding free energies was carried out on shorter time windows in order to get three replicates but all three segments were taken from the same simulation time (800-1500 ns). In many publications, this kind of calculation is performed with smaller windows (5 or 10 ns, likely due to the computational cost of the simulations) leading to less accurate results. We also tried to compute these energies using the full-length trajectory but the results were unsatisfactory with large standard deviation.
- The sentence “we performed MM/GBSA calculations in triplicate on segments of 100-200 ns of the simulation (in the range from 800 ns to 1500 ns) with implicit solvation model” is ambiguous.
Reformulate the sentence “we performed free binding energy calculation ….with MM/GBSA as implicit solvation model or “the GBx implicit solvation model
We thank the reviewer for pointing out this mistake and the sentence has been reformulated.
- I do not fully understand the meaning of “blue: Q; green: R; and red: K” in Figure 3.
Each color corresponds to the residue in position 493 which is variable across variants (Gln in WT, Arg (in BA.1 and BA.2 variants) or Lys for BA.1+Q493K. We applied these colors to quantify and illustrate the non-bonded energy of this residue alone at position 493 with all residues from hACE2 (top-colored curves in fig. 3). We have modified the legend of the Fig.3 more clarity.
- Minor Issues:
Please pay attention to small grammatical errors, such as “Total energy contributions per residues computed.
We have corrected the sentence in the legend of figure 6.
Reviewer 2 Report
Comments and Suggestions for Authors
This MD study of SARS-CoV-2 written by Aria Gheeraert et. al. investigates the impact of mutations in SARS-CoV-2 variants on the binding interactions between the receptor-binding domain (RBD) and human ACE2 (hACE2). The work is well-structured and uses an array of computational approaches, such as PairInt energy calculations, MM/GBSA free energy estimation, and PCA analysis, providing insights into how mutations in SARS-CoV-2 variants impact RBD-hACE2 binding. The findings suggest that Omicron variants exhibit different interaction profiles compared to Delta, likely contributing to their enhanced transmissibility. In addition, considering the damage the virus made to our human beining and its potential damage in the future,I think it is worth to study the virus still. Before publication, the author should answer my following concerns.
- Many study about RBD binding combined both computational work and experimental method, such as the study of N501Y and S373P effect. Nevertheless, I think it is OK as a pure computational work for RBD binding, which still provide important molecular/atomic information. However, the computational findings should be cross-referenced with published experimental studies, such as those using SPR or FACS to measure binding affinities. The author did a good job on the structural information, they should also did this on the binding affinity.
-
In addition, the limitation of computional methods should be mentioned, especially for this pure MD study. For example, the accuracy of the force field, some solvent effect, etc.
- MD can give a good result about the binding affinity result. But the viral transmission rate is not directly dependent on the RBD binding affinity. the authour should disscuss about it.
Author Response
In this study, the authors use molecular dynamics (MD) simulations of the spike RBD/hACE2 complex of SARS-CoV-2 variants (WT, Delta, and Omicron) to gain insight into the spike affinity for the human ACE2 63 (hACE2) target that may be associated with viral infection. In principle, the study is interesting and can be considered for publication in Biomolecules. My only comment is that many figures (e.g. 3 & 4) are of very low quality and should be revised. In addition, some 3D structures could be presented in a more elaborate way.
We thank the reviewer for the general appreciation of our work. We have improved the quality of Figures 3 and 4 as the resolution was unsatisfactory (the resolution was much lower in the pdf file than in the Word document). We also have modified the panel D in Figure 9 to better represent the complex with the residues having high eigenvector centrality values.
Reviewer 3 Report
Comments and Suggestions for Authors
In this study, the authors use molecular dynamics (MD) simulations of the spike RBD/hACE2 complex of SARS-CoV-2 variants (WT, Delta, and Omicron) to gain insight into the spike affinity for the human ACE2 63 (hACE2) target that may be associated with viral infection. In principle, the study is interesting and can be considered for publication in Biomolecules. My only comment is that many figures (e.g. 3 & 4) are of very low quality and should be revised. In addition, some 3D structures could be presented in a more elaborate way.
Author Response
This MD study of SARS-CoV-2 written by Aria Gheeraert et al. investigates the impact of mutations in SARS-CoV-2 variants on the binding interactions between the receptor-binding domain (RBD) and human ACE2 (hACE2). The work is well-structured and uses an array of computational approaches, such as PairInt energy calculations, MM/GBSA free energy estimation, and PCA analysis, providing insights into how mutations in SARS-CoV-2 variants impact RBD-hACE2 binding. The findings suggest that Omicron variants exhibit different interaction profiles compared to Delta, likely contributing to their enhanced transmissibility. In addition, considering the damage the virus made to our human being and its potential damage in the future, I think it is worth to study the virus still. Before publication, the author should answer my following concerns.
- Many study about RBD binding combined both computational work and experimental method, such as the study of N501Y and S373P effect. Nevertheless, I think it is OK as a pure computational work for RBD binding, which still provide important molecular/atomic information. However, the computational findings should be cross-referenced with published experimental studies, such as those using SPR or FACS to measure binding affinities. The author did a good job on the structural information, they should also did this on the binding affinity.
We agree with the reviewer’s comments concerning the comparison between theoretical and experimental data, we tried to include many published works but it is impossible to mention all publications. For this reason, we decided to include a table (Supp. Table S5) showing a non-exhaustive list of publications reporting binding affinity measurements and the data obtained on binding affinities have been discussed in the conclusion.
- In addition, the limitation of computational methods should be mentioned, especially for this pure MD study. For example, the accuracy of the force field, some solvent effect, etc.
A sentence has been included in the final section to inform and raise awareness among readers about these sensitive parameters used in MD simulations (initial coordinate system, force field parameters, duration of simulation, water model type…).
- MD can give a good result about the binding affinity result. But the viral transmission rate is not directly dependent on the RBD binding affinity. The author should discuss about it.
We tried to discuss this particular point although the task is difficult and complex. Indeed, we cannot directly correlate the binding affinity to the high transmission rate or the infectiousness since the binding step is followed by many other steps. A strong binding is required to allow the conformational change and thereafter the cleavage of Spike that will lead to the release of SARS-CoV-2 genetic material inside the host cell. However, the immune system also plays a major role in the viral transmission rate by preventing or by slowing down theses early steps. The message here, would like to insist more on the plasticity promoted by Spike mutations, giving the final variant an advantage respecting a balance between immune system escape and a strong binding to hACE2. This point was already largely discussed in the conclusion.
Round 2
Reviewer 1 Report
Comments and Suggestions for Authors
All the concerns are solved.
Reviewer 2 Report
Comments and Suggestions for Authors
the authors have answered all my questions